



# The blessing of dimensionality for the analysis of climate data

Bo Christiansen

Danish Meteorological Institute, Copenhagen, Denmark

**Correspondence:** Bo Christiansen (boc@dmi.dk)

**Abstract.** We give a simple description of the blessing of dimensionality with the main focus on the concentration phenomena. These phenomena imply that in high dimensions the length of independent random vectors from the same distribution have almost the same length and that independent vectors are almost orthogonal. In climate and atmospheric sciences we rely increasingly on ensemble modelling and face the challenge of analysing large samples of long time-series and spatially extended

fields. We show how the properties of high dimensions allow us to obtain analytical results for, e.g., correlations between sample members and the behaviour of the sample mean when the size of the sample grows. We find that the properties of high dimensionality with reasonable success can be applied to climate data. This is the case although most climate data show strong anisotropy and both spatial and temporal dependence resulting in effective dimensions around 25-100.

## 1  Introduction

In many areas of geophysics we operate in high-dimensional spaces. Examples from the atmospheric and climate sciences include extended spatial fields, such as precipitation or near-surface temperature, and long time-series of atmospheric variables, such as the global mean temperature. These fields and time-series may be either observed or modelled. Over the last decades ensemble modelling has been generally accepted as a valuable tool to gauge the unpredictability and error originating from uncertain initial conditions or deficiencies in model physics. There is also an increased tendency for gridded observational

products and reanalyses to apply ensemble techniques to represent the different uncertainties. We are therefore often in a situation where we need to analyse large samples of high dimensional fields. These samples could consist of the individual ensemble members or just individual years of a spatial field.

This might seems a daunting challenge as the properties of high dimensional space often appear counter-intuitive to minds experienced only in the low dimensional world. However, the properties of high dimensional space may sometimes simplify the

analysis and allow us to obtain rather general analytical results. A central result in this respect is the concentration of measures which, with a quote from Chazottes (2015), loosely states: *A random variable that smoothly depends on the influence of many weakly dependent random variables is, on an appropriate scale, very close to a constant.* The importance of this is described with another quote: *The idea of concentration of measures is arguably one of the great ideas of analysis in our time* (Talagrand, 1996). We will see how in high dimensions such concentration properties often allow us to substitute the length of a random

vector with its expectation value and to treat independent vectors as orthogonal (i.e., having zero dot product).





These advantageous properties of high dimensionality – often referred to as the blessing of dimensionality – have rarely been applied to the atmospheric and climate sciences. Exceptions are our previous papers on the subject. In Christiansen (2018) we described how the blessing of dimensionality explains why the ensemble mean often outperforms the individual ensemble members and why the ensemble mean often has an error that is 30 % smaller than the median error of the individual

ensemble members. In Christiansen (2019) we used the properties of high dimensions to analyse a global ensemble reforecast. We described how the behaviour of the ensemble mean forecast can be described by a simple model in which variances and bias depend on lead-time. In Christiansen (2020) we analyzed a multi-model climate ensemble using the properties of high dimensions to separate two competing understandings of the ensemble – the indistinguishable interpretation and the truth centered interpretation. In this paper we aim to give a more comprehensive and coherent discussion of the blessing of

dimensionality and to which extent it applies to the situation in atmospheric science.

In section 2 we describe the properties of high dimensional spaces focusing first on what is often called the curse/blessing of dimensionality (subsection 2.1) and then more specifically on the concentration of measures (subsection 2.2). The mathematical results are often only proved for independent and identically distributed (iid) random variables. In section 3 we discuss how this requirement can be loosened and how it relates to geophysical fields which often contains strong temporal and spatial

dependence. In section 4 we focus on the application to atmospheric and climate science. First, in subsection 4.1 we directly investigate to which extent the climatic fields fulfill the requirements of high dimensionality. We then (subsections 4.2 and 4.3) discuss analytical results for distances and correlations between samples and how well these hold for climate fields. In subsection 4.4 we likewise explore analytical results for how the ensemble mean depends on ensemble size. The paper is closed with the conclusions in section 5.

**2   Properties of high dimensional spaces**

Here we give a brief overview of the properties of high dimensional spaces. We begin in subsection 2.1 with some general considerations about high dimensional spaces while we in subsection 2.2 focus more on the concentration of measures. Some of the simple examples were also, but more briefly, described in Christiansen (2018).

**2.1   Curse of dimensionality**

The properties of high dimensional spaces often defy our intuition based on two and three dimensions (Cherkassky and Mulier, 2007; Bishop, 2007; Blum et al., 2020). Apart from the well-known fact – sometimes called the empty space phenomenon – that the number of samples needed to obtain a given coverage grows exponentially with dimension, there are other less appreciated features of high dimensional spaces (Blum et al., 2020). For example, almost every point is an outlier in its own projection and independent vectors are almost always orthogonal. The latter property is called waist concentration and, more precisely, states

that when the dimension increases the angles between independent vectors become narrowly distributed around the mean $\pi/2$ with a variance that converges towards zero.





**Table 1.** Results for a unit cube in N dimensions. The vertices of a unit cube $[-1/2, 1/2]^N$ are $[\pm 1/2, \pm 1/2, \dots \pm 1/2]$. The number of vertices is $2^N$ and the length of the vertices $\sqrt{N}/2$. The fraction of volume within $\epsilon$ of the edge is $1 - (1 - \epsilon)^N$. The volume of inscribed sphere is $\pi^{N/2}(d/4)^N/\Gamma(N/2 + 1)$ with $d = 1$.

| N | Volume | # vertices | Length of vertices | Volume of inscribed sphere | Fraction of volume within 0.05 of edge |
|---|---|---|---|---|---|
| 2 | 1 | 4 | 0.707 | 0.785 | 0.0975 |
| 3 | 1 | 8 | 0.866 | 0.524 | 0.1426 |
| 5 | 1 | 32 | 1.118 | 0.164 | 0.2262 |
| 10 | 1 | 1024 | 1.581 | 0.00249 | 0.4013 |
| 25 | 1 | $3.35\ 10^7$ | 2.500 | $2.85\ 10^{-11}$ | 0.7226 |
| 50 | 1 | $1.13\ 10^{15}$ | 3.535 | $1.54\ 10^{-28}$ | 0.9231 |
| 100 | 1 | $1.27\ 10^{30}$ | 5.000 | $1.87\ 10^{-70}$ | 0.9941 |

The properties of high dimensional spaces are sometimes called the curse and sometimes the blessing of dimensionality depending on the considered problem. In the present context these properties turn out to be a blessing as they strongly simplify the analysis and make analytical results possible. The beneficial properties of high dimensionality are recognized in many areas of machine learning (Kainen, 1997; Gorban and Tyukin, 2018).

As a simple example we consider a cube in $N$ dimensions with side $d$ and centered around **0**. The cube has $2^N$ vertices with the positions $d(\pm 1/2, \pm 1/2, \dots \pm 1/2)$. The distance between each vertex and the center is $d\sqrt{N}/2$. The volume of the cube within a distance $\epsilon d$ of the edge is $(d^N - (d - \epsilon d)^N)/d^N = 1 - (1 - \epsilon)^N$ and the volume of the inscribed sphere is $\pi^{N/2}(d/4)^N/\Gamma(N/2 + 1)$. The situation is shown in Table 1 for a unit cube ($d = 1$) for different values of $N$. For $N = 100$ there are more than $10^{30}$ vertices [1] and more that 99 % of the volume is within a distance 0.05 of the edge. The volume of the inscribed sphere – which for 2 dimensions contains the bulk of the cube – is virtually zero. Thus, the volume increasingly concentrates near the surface when the dimension increases. The form of the $N$-dimensional cube has been compared to that of a sea urchin (Hecht-Nielsen, 1990).

Consider now a sample of points drawn independently from the high dimensional cube. For moderate sample size ($\ll 2^N$, which already for $N = 25$ is larger than $10^7$, so moderate is probably not the right word) all samples will be located in different vertices. This means that all samples will have almost the same distance from the center and that all pairs of samples will be almost perpendicular. The distances between pairs of samples will also be almost identical making concepts such as nearest neighbours problematic. These properties are not particular for the cube, but are quite general also for unbounded distributions as we will see in the next subsection.

---

[1] Comparable to the number of atoms in 30 tons of water or the number of bacteria on the Earth. A factor of one million larger than the estimated number of stars in the universe.


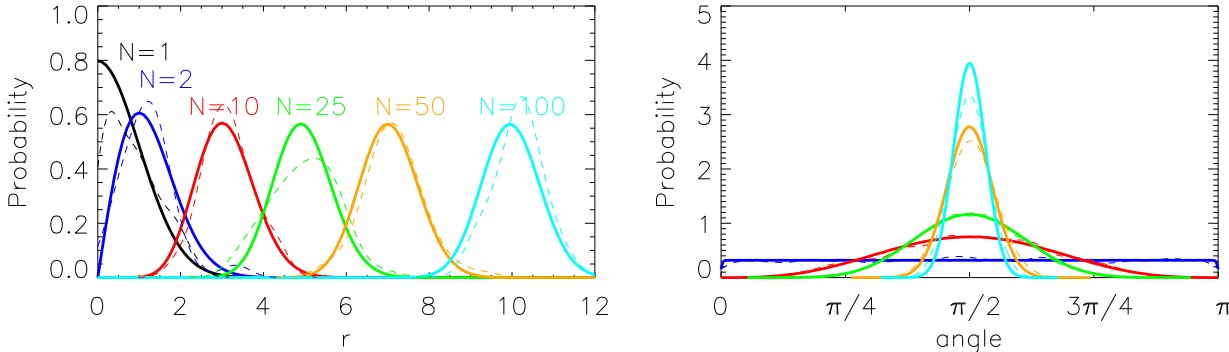

**Figure 1.** Left: $N$-dimensional Gaussian distributions with unit variances and zero means as function of $r = ||\mathbf{x}||$ for different values of $N$ (Eq. 3). The position of the mode goes like $\sqrt{N}$ and the width is approximately the constant $1/\sqrt{2}$. Left: The distribution of angles between pairs of independent $N$-dimensional Gaussian vectors for the same values of $N$. Thick curves are calculated in the large ensemble limit. For both left and right panels the thin dashed curves illustrate the distributions for a sample of size 50.

## 2.2 Concentration of measures

We first look at a very simple example to describe the general idea of concentration of measures. Consider $N$ iid random variables $x_i$, $i = 1, 2 \ldots N$, each with mean $\mu$ and variance $\sigma^2$. For the sum of the variables, $\sum_i x_i$, we have for the expectation and variance:

$$E(\sum_i x_i) = \sum_i E(x_i) = N\mu \tag{1}$$

and

$$Var(\sum_i x_i) = \sum_i Var(x_i) = N\sigma^2. \tag{2}$$

Thus, $\sqrt{Var(\sum_i x_i)}/E(\sum_i x_i) = \sqrt{\frac{1}{N}}\frac{\mu}{\sigma}$. Therefore, when $N$ grows both the expectation of the sum and the width of its distribution will grow, but the relative width will decrease. We can therefore, with some reason, say that the distribution of the sum becomes more and more sharply defined around its mean. If we normalize the sum with $N$ to get the mean, $\overline{x} = \sum_i x_i/N$, we have $E(\overline{x}) = \mu$ and $Var(\overline{x}) = \sigma^2/N$. Therefore, the mean becomes increasingly narrowly distributed around the constant $\mu$. Thus, for large $N$ we can in many situations treat the mean, $\overline{x}$, as a constant.

The considerations above are basically the rationale behind the law of large numbers and are also closely related to the central limit theorem which states that $(\overline{x} - \mu)/\sqrt{N}/\sigma$ converges towards a standard Gaussian distribution, $\mathcal{N}(0, 1)$. The concentration of measures can be extended beyond the iid situation (see section 3) as indicated by the quotation from Chazottes (2015) in the introduction.

Let us organize the random variables into an N-vector $\mathbf{x} = (x_1, x_2 \ldots x_N)$. Now $||\mathbf{x}||^2 = x_1^2 + x_2^2 \ldots x_N^2$ is a sum of independent variables and will therefore – according to the arguments above – for large $N$ be approximately a constant.





Let us consider a multi-variate standard Gaussian distribution $P(\mathbf{x}) = (2\pi)^{-N/2} \exp(-\sum_{n=1}^{N} x_n^2/2)$. The surface area of a hyper-sphere with radius $r$ in $N$ dimensions is $S_{N-1} = 2\pi^{N/2} r^{N-1}/\Gamma(\frac{N}{2})$. So as function of $r = ||\mathbf{x}||$ we get the $\chi$-distribution:

$$P(r) = S_{N-1} P(\mathbf{x}) = \frac{2^{1-N/2}}{\Gamma(\frac{N}{2})} r^{N-1} \exp(-r^2/2). \tag{3}$$

The maximum of $P(r)$ is reached for $r = \sqrt{N-1}$ and the width (standard deviation) of the peak converges fast with $N$ towards $1/\sqrt{2}$. This is illustrated in Fig. 1.

The concentration of measures is the backbone of statistical mechanics. As a simple example, we consider the canonical ensemble of weakly interacting identical particles. This ensemble describes a system with a constant number of particles, $N$, in a heat-bath. All particles have the same spectrum of energy states, $E_i$, and the probability for a particle to be in the $i$th state is proportional to $\exp(-\beta E_i)$. The total energy grows like $N$, while the fluctuations (standard deviation) in the total energy grows like $\sqrt{N}$. Thus, the relative fluctuations in the total energy go as $\sim 1/\sqrt{N}$ and in the thermodynamic limit, $N \to \infty$, these fluctuations and fluctuations in other macroscopic quantities can be neglected. This holds also for non-identical and interacting particles (see Gorban and Tyukin, 2018, for a recent discussion) just as the concentration of measures can be extended beyond the iid situation.

Let us take a brief look at waist concentration. Consider two independent unit vectors $\mathbf{a}$ and $\mathbf{b}$. Without lack of generality we can set $\mathbf{a} = (1, 0, 0 \ldots)$. The dot-product then becomes $b_1$. It is therefore easy to see that $\mathbf{a} \cdot \mathbf{b}$ has zero mean and that its spread converges to zero as $\sim 1/\sqrt{N}$. The angle $\phi$ between $\mathbf{a}$ and $\mathbf{b}$ will therefore converge towards $\pi/2$ as $\cos\phi = \mathbf{a} \cdot \mathbf{b}$. This is illustrated in Fig. 1 for Gaussian distributed vectors for different values of $N$: For $N = 2$ the distribution of angles is flat, but for larger values of $N$ it becomes increasingly peaked around $\pi/2$.

The topic of concentration properties is an active mathematical field with focus on probabilistic bounds on how fast empirical means converge to the ensemble means for different classes of random variables including non-iid variables (Vershynin, 2018; Wainwright, 2019). Such bounds include Bernstein's and Hoeffding's inequalities and give strict mathematical meaning to the looser considerations above. As an example the Hoeffding bound states that for all $t \geq 0$:

$$P\left( \left| \frac{1}{N} \sum (x_i - \mu_i) \right| \geq t \right) \leq \exp(-Nt^2/(2\sigma^2)). \tag{4}$$

This holds for sub-Gaussian independent random variables (Wainwright, 2019). Here $\mu_i$ is the mean of $x_i$ and $\sigma$ is a constant. For the angle $\phi$ between two independent vectors we have similarly for all $t \geq 0$ (from Gorban and Tyukin, 2018):

$$P(|\cos\phi| \geq t) \leq 2 \exp(-Nt^2/2). \tag{5}$$

## 3 Extension to situations with dependent and non-identical variables

Like the central limit theorem (CLT), the concentration properties are originally developed for iid variables. However, also as the central limit theorem, they can be extended to classes of dependent variables. Although no general condition exists





for the CLT (Clusel and Bertin, 2008), an important factor for both the CLT and the concentration properties is the strength of the dependence (Kontorovich and Ramanan, 2008; Chazottes, 2015). Many properties of iid processes can be extended to

processes where the rate of mixing is strong enough (Chazottes, 2015). Here, mixing processes are defined by a decay of correlations towards zero, i.e., $x_i$ and $x_j$, should become independent when $|i - j|$ increases.

Here correlations generally refer to measures of the dependence, e.g., the distance between the joint distribution and the product of the marginal distributions. Note, that the decay of Pearson's correlation coefficient is not necessarily sufficient as a zero correlation coefficient does not guarantee independence as it only gauges linear dependence. The auto-regressive moving-

average (ARMA) models which are often used in geophysics are examples of mixing processes (Mokkadem, 1988). More generally Chazottes (2015) finds that the concentration of measures holds for a random variable that smoothly depends on the influence of many weakly dependent random variables.

The mixing and decay of correlations are closely related to the concept of effective degrees of freedom also known as the effective dimension (e.g., Clusel and Bertin, 2008). Shalizi 2006 [2] shows an example of the CLT for dependent variables '..

only with the true sample size replaced by an effective sample size ..' The basic idea is that dependent variables of effective dimension $N^*$ gives the same information as independent variables of dimension $N$. Heuristically, consider a function on a 2-dimensional square region with each side of length $L$. If correlations decay exponentially with characteristic length $\xi$, then we can to a first approximation describe the function by $N^* = (L/\xi)^2$ independent variables. For fixed $\xi$ the number of independent variables go to infinity with increasing $L$ and in this situation we may assume that the limit theorems hold. Note,

that some methods to calculate the number of effective dimensions of, e.g., surface temperature are directly based on these arguments using an average $\xi$ (see the summary in Christiansen and Ljungqvist, 2017).

The situation is well-known in the study of 1-dimensional time-series (see, e.g., von Storch and Zwiers, 1999, section 17). As a simple example we consider a time-series, $x_i$, of length $N$ generated with a first order auto-regressive, AR(1), process with coefficient $\rho$. The auto-correlations behave as $\sim \rho^r$, where $r$ is the lag. A decorrelation time, $\tau$, can be found where the

auto-correlations have decayed to $e^{-1}$: $\tau = -1/\ln\rho$. The effective degrees of freedom would therefore be $-N\ln\rho$. Less heuristically, we have for the ensemble mean, $\bar{x} = \sum_i x_i/N$, that $\sqrt{N}\bar{x} \sim \mathcal{N}(0, \sigma^2(1+\rho)/(1-\rho))$. Comparing with the similar results for iid Gaussian variables, $\sqrt{N}\bar{x} \sim \mathcal{N}(0, \sigma^2)$, this suggests an effective dimension of $N(1-\rho)/(1+\rho)$ (Crack and Ledoit, 2009; von Storch and Zwiers, 1999). For the sum of squares we have $\sqrt{N}\bar{x^2} \sim \mathcal{N}(\sigma^2, 2\sigma^4(1+\rho^2)/(1-\rho^2))$, now suggesting an effective dimension of $N(1-\rho^2)/(1+\rho^2)$ (Bartlett, 1935). We note, that the effective dimension $N^*$ depends on the

measure of interest.

In the case of 2-dimensional fields different methods exist to estimate the number of effective dimensions $N^*$ (Wang and Shen, 1999; Bretherton et al., 1999). Some methods are directly based on the characteristic length, $\xi$, using an average over the different directions (Christiansen and Ljungqvist, 2017). The estimated number depends both on the method used and on the field, the time-scale, and the geographical region. For annual mean surface-temperatures values of $N^*$ vary between 50 and 100

depending on method (Briffa and Jones, 1993; Hansen and Lebedeff, 1987; Shen et al., 1994) when the whole globe is consid-

---

[2]https://www.stat.cmu.edu/~cshalizi/754/2006/notes/lecture-27.pdf

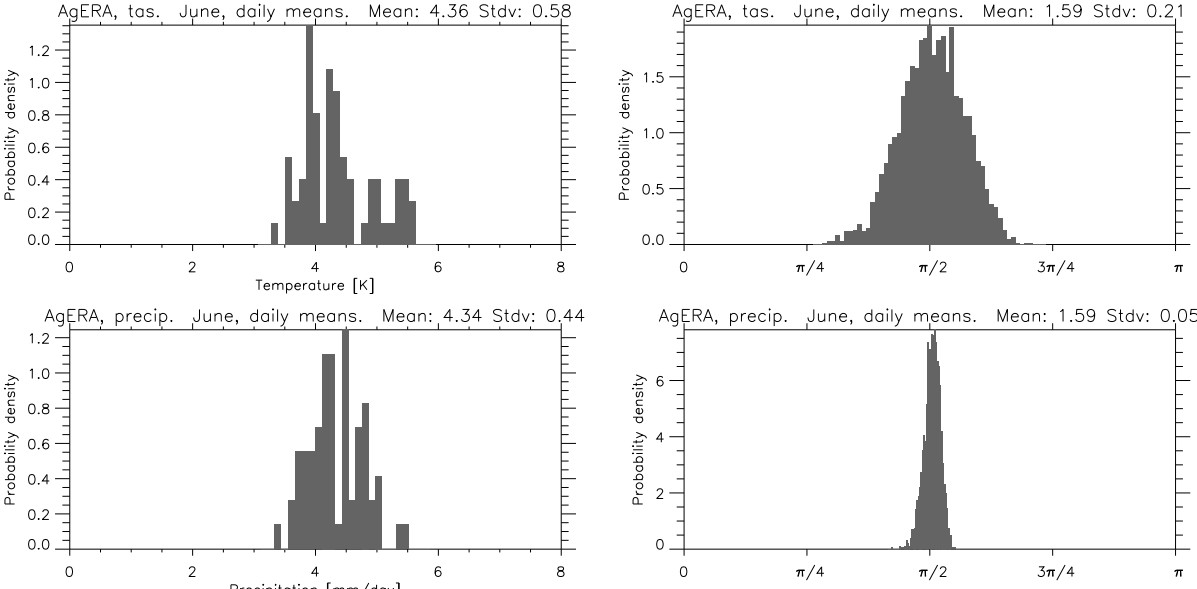

**Figure 2.** The AgERA data-set. Daily means with five days separation for June, 1980-1990. Left: distribution of the lengths. Right panel: distribution of the angles. Top: Near surface temperature [K]. Bottom: Precipitation [mm/day].

ered. Values in the same range have been found for monthly surface temperatures in the northern hemisphere (Wang and Shen, 1999; Bretherton et al., 1999).

These numbers are off course small compared to Avogadro's number relevant for statistical mechanics, but they are still comparable to the dimensions in Fig. 1 where the concentration properties hold to a reasonable degree. In subsection 4.b we

directly investigate to which extent the concentration properties hold for atmospheric fields.

## 4    Atmospheric and climate science

As we saw in section 2, concentration of measures and waist concentration allow us in high dimensions to set dot products of independent vectors to zero and substitute the length of a random vector with its expectation value. In section 3 we argued that when the components of the fields or time-series are dependent the concentration phenomena hold when the effective

dimension is large. However, to test the concentration properties we also need independent samples.

For initial condition ensembles consisting of experiments with the same model but with different initial conditions, the different ensemble members can be considered independent (considering anomalies with respect to the ensemble center as explained in the next subsection). For multi-model ensembles where experiments are performed with models with different physical parameterizations (but the same external forcings) the situation is more complicated (e.g., Knutti et al., 2013; Boé,

2018; Christiansen, 2020, and references therein). The annual or monthly climatologies are obvious measures for comparing





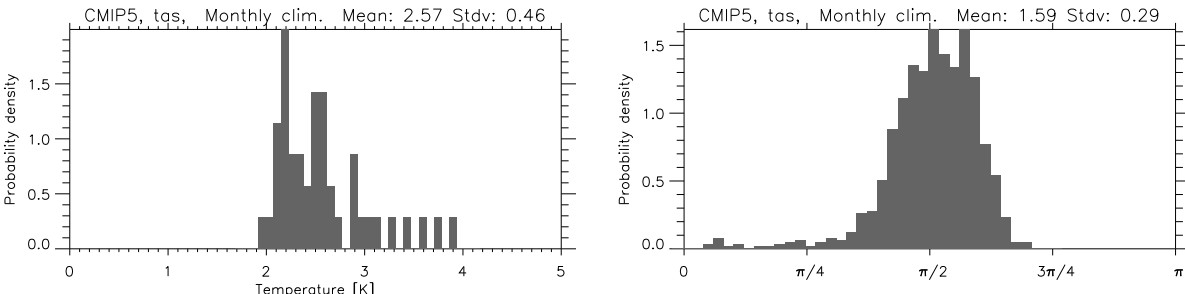

**Figure 3.** The multi-model 45 member CMIP5 ensemble. Monthly climatology in TAS [K]. Left panel: distribution of the lengths. Right panel: distribution of the angles.

models or for validating the models against observations (Gleckler et al., 2008). Other used measures are the forced response in, e.g., time-series of global means.

Another way to obtain independent samples from the same distribution is to consider a given variable at different times. For example, we could look at the spatial field of precipitation or temperature at different days or months. To ensure that the fields
are drawn from the same distribution we need to avoid or remove the annual cycle and – if longer periods are considered – to make sure that there is no external forcing. The sample times should also be sufficiently separated.

In the next sections we will consider the following geophysical data-sets; 1) Daily means of near-surface temperature and precipitation from AgERA for June in the period 1980-1990. The AgERA provides daily surface meteorological data for agro-ecological studies (doi: 10.24381/cds.6c68c9bb) based on ECMWF's ERA5 reanalysis (Hersbach et al., 2019). AgERA is
land-only and of high resolution with more than two million ($N = 2353526$) grid-points. 2) Monthly near-surface temperature from the multi-model CMIP5 ensemble (Flato et al., 2013) consisting of 45 historical experiments. The models are identified in Table 1 of Christiansen (2020). 3) The Max Planck Institute Grand Ensemble (MPI-GE, Maher et al., 2019) consisting of 100 members differing only in initial conditions. From MPI-GE we consider the monthly mean near-surface temperature and precipitation. For both model ensembles we consider the monthly climatology in the period 1980-2005 and the annual Northern
Hemisphere (NH) mean values in the period 1961–2005.

In addition to the geophysical data we also include two simple samples of independent vectors. The first sample consists of independent vectors drawn from an $N$-dimensional spherical (all components have zero mean and unit variance) Gaussian distribution as in Fig. 1. The second sample is drawn from a standard Gamma distribution with shape parameter 3 (location and scale parameters 0 and 1). In the latter case we include anisotropy (not identically distributed components) by multiplying
the $n$th component with $5n/N$, so the mean and variance of the $n$th component become $15n/N$ and $75(n/N)^2$, respectively. For the simple random variables we let the dimension, $N$, vary from 1 to 100. The sample size is chosen to 50.

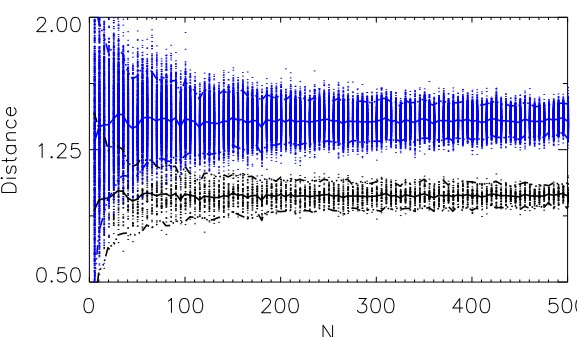
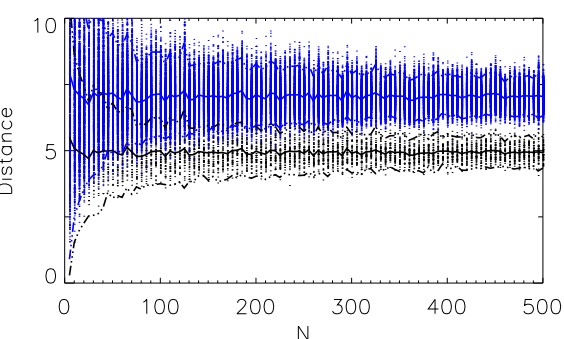

**Figure 4.** The normalized distances as function of dimension $N$. For each $N$ we draw 50 $N$-dimensional random vectors and calculate the pair-wise distances $\sqrt{||\mathbf{x}^k - \mathbf{x}^l||^2/N}$ (blue) and the distances to the sample mean $\sqrt{||\mathbf{x}^k - \overline{\mathbf{x}}||^2/N}$ (black). The thick curves show the mean of the distances, and the broken curves the mean $\pm$ two standard deviations. In left panel each component of the vectors is drawn from a standard Gaussian, in right panel each component is drawn from a standard Gamma distribution with shape parameter 3 (location and scale parameters 0 and 1). In the latter case we include anisotropy by multiplying the $n$th component with $5n/N$, so the mean and variance of the $n$th component become $15n/N$ and $75(n/N)^2$. Note the factor of $\sqrt{2}$ between the mean distances.

### 4.1 Concentration of measures in atmospheric fields

In this subsection we directly investigate the distributions of the lengths of the sample members and the distributions of the angles between them. The results from this and the following subsections are summarized in Table 2.

We center the sample, $\mathbf{x}^k$, $k = 1, \ldots K$, to the sample mean, $\overline{\mathbf{x}} = \sum_k \mathbf{x}^k / K$, and calculate the lengths as the square root of $||\mathbf{x}^k - \overline{\mathbf{x}}||^2/N$ for each sample member. The angle $\phi$ between two sample members, $k$ and $l$, is given by $(\mathbf{x}^k - \overline{\mathbf{x}}) \cdot (\mathbf{x}^l - \overline{\mathbf{x}}) = ||\mathbf{x}^k - \overline{\mathbf{x}}|| \, ||\mathbf{x}^l - \overline{\mathbf{x}}|| \cos \phi$. This gives us $K$ lengths and $K(K-1)/2$ angles. This centering – the subtraction of the sample mean – is not important for the calculation of the lengths as we explain in the end of this subsection.

We first consider the near surface temperature and precipitation fields from the AgERA data-set. Figure 2 shows the lengths and angles for daily means taken every fifth day for June in the period 1980-1990. The 11 years gives us 66 samples. We see that for temperature the lengths are relatively tightly distributed around 4.36 K ($\sigma$ in Eq. 6) with a standard deviation of 0.58 K. The angles are likewise distributed around $\pi/2$ with a standard deviation of 0.21. For precipitation the distributions are somewhat narrower in particular for the angles. This is what we would expect due to the larger number of effective degrees of freedom compared to temperature. However, this effect is reduced as we include both dry and wet days in the analysis. While the precipitation amount on wet days has a short decorrelation length this does not hold for the spatial field indicating wet/dry days. Note also that the distribution of precipitation is extremely non-Gaussian. These results indicate that the concentration of measures and the waist concentration hold at least to some extent for these field.

Figure 3 shows the lengths and angles for the monthly seasonal cycle in near-surface temperature, 1980-2015, for the multi-model CMIP5 ensemble. The models have been regridded to a common 144x73 grid so N = 144x73x12. The sample has a size of 45 and consists of one ensemble member from each of the models. The lengths are distributed around 2.57 K with a standard





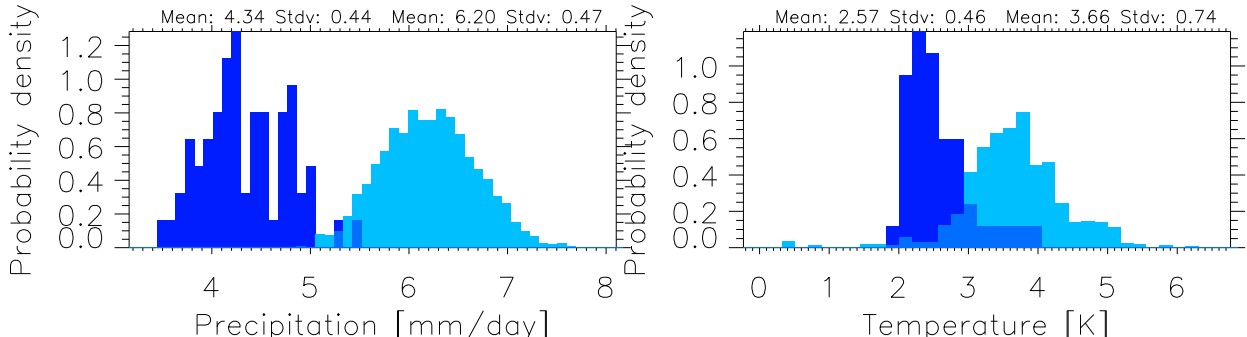

**Figure 5.** Distances between samples (cyan) and between sample and sample mean (blue). Left: AgERA, daily mean precipitation for June. Right: CMIP5, monthly climatology of near-surface temperature. Note the factor of $\sqrt{2}$ between mean distances.

deviation of 0.46 K and the angles around $\pi/2$ with a standard deviation of 0.28. Thus, compared to the example in Fig. 2 the distributions are less tightly distributed. The main explanation is probably that the effective degrees of freedom in the monthly climatology is smaller than that of the daily fields. However, there are also reasons to believe that the multi-model ensemble is not totally independent (Knutti et al., 2013; Boé, 2018). Note, the negative skewness in the distribution of the angles. Angles

close to zero indicate pairs of models that are almost parallel and therefore strongly dependent. These pairs correspond to variants of the same model, such as MIROC-ESM and MIROC-ESM-CHEM, which are well known to be close in the model genealogy (Knutti et al., 2013).

Results for the MPI-GE 100 member initial condition ensemble are shown in Table 2. Here we have 192x96 grid-points so N = 192x96x12 and the sample size is 100. The distributions of lengths and angles are now narrower compared to the multi-

model CMIP5 ensemble. This corresponds to a larger effective dimension in the monthly climatology which now reflects only different initial conditions and not model differences. Also in this example are the distributions for precipitation narrower than those for temperature.

Reducing the spatial area decreases the effective dimension. As an example we have included in Table 2 the results for the AgERA when applied to Northern Europe (50–65°N, 0–25°E). As expected we see an increase in the width of the distributions

for both precipitation and near surface temperature.

In the analysis above we centered the sample to the sample mean before calculating the lengths, i.e., we used $||\mathbf{x}^k - \overline{\mathbf{x}}||^2/N$ instead of $||\mathbf{x}^k||^2/N$. But these expressions only differ by the length of the mean: $||\mathbf{x}^k||^2/N = \overline{\mathbf{x}}^2/N + ||\mathbf{x}^k - \overline{\mathbf{x}}||^2/N$ as $\mathbf{x}^k - \overline{\mathbf{x}}$ and $\overline{\mathbf{x}}$ are orthogonal in high dimensions. The absence of centering makes most sense for precipitation that has a natural zero point. For AgERA precipitation $||\overline{\mathbf{x}}||/\sqrt{N}$ is 3.35, the mean of $||\mathbf{x}^k - \overline{\mathbf{x}}||/\sqrt{N}$ is 4.34, and the mean of $||\mathbf{x}^k||\sqrt{N}$ is 5.48 (all

mm/day), fulfilling the Pythagorian relationship ($5.48^2 = 3.35^2 + 4.34^2$).





**Table 2.** Summary of the different measures. Entries show mean/standard deviation. Units are K for temperature and mm/day for precipitation.

| | Lengths | Angles | Distances between pairs of ensemble members | Distances between ensemble members and ensemble mean | Correlations between pairs of ensemble members | Correlations between ensemble members and ensemble mean |
|---|---|---|---|---|---|---|
| AgERA, temperature, daily, June | 4.36/0.58 | 1.59/0.21 | 6.21/0.88 | 4.36/0.58 | 0.41/0.13 | 0.64/0.07 |
| AgERA, precipitation, daily, June | 4.34/0.44 | 1.59/0.05 | 6.20/0.47 | 4.34/0.44 | 0.46/0.05 | 0.69/0.04 |
| AgERA, temperature., daily, June, N. Europe | 3.25/1.06 | 1.58/0.50 | 4.61/1.58 | 3.25/1.06 | 0.35/0.31 | 0.57/0.25 |
| AgERA, precipitation, daily, June, N. Europe | 4.05/1.88 | 1.56/0.28 | 5.92/2.32 | 4.95/1.88 | 0.66/0.17 | 0.82/0.13 |
| CMIP5, monthly climatology, temperature | 2.57/0.46 | 1.59/0.28 | 3.66/0.73 | 2.57/0.46 | 0.44/0.18 | 0.64/0.12 |
| MPI-GE, monthly climatology, temperature | 0.34/0.03 | 1.58/0.12 | 0.49/0.04 | 0.34/0.03 | 0.49/0.07 | 0.70/0.04 |
| MPI-GE, monthly climatology, precipitation | 0.27/0.01 | 1.58/0.06 | 0.39/0.02 | 0.27/0.01 | 0.51/0.04 | 0.72/0.03 |
| CMIP5, NH annual means, temperature | 0.67/0.35 | 1.60/1.14 | 0.90/0.59 | 0.67/0.35 | 0.39/0.19 | 0.63/0.10 |
| MPI-GE, NH annual means, temperature | 0.16/0.02 | 1.58/0.20 | 0.22/0.03 | 0.16/0.02 | 0.54/0.12 | 0.74/0.07 |

## 4.2 Distances between samples and between samples and ensemble mean

If the sample members are drawn independently from the same distribution in high dimensions they have approximately the same length, and we can write

$$||\mathbf{x}^k - \overline{\mathbf{x}}||^2/N = \sigma^2. \tag{6}$$

For the distance between two different sample members we get:

$$||\mathbf{x}^k - \mathbf{x}^l||^2/N = ||(\mathbf{x}^k - \overline{\mathbf{x}}) - (\mathbf{x}^l - \overline{\mathbf{x}})||^2/N = 2\sigma^2, \tag{7}$$

where we have used that $\mathbf{x}^k - \overline{\mathbf{x}}$ and $\mathbf{x}^l - \overline{\mathbf{x}}$ are orthogonal.

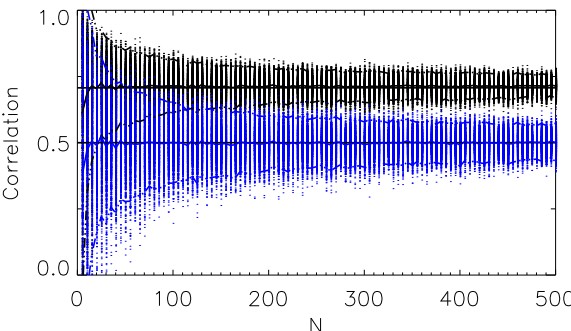
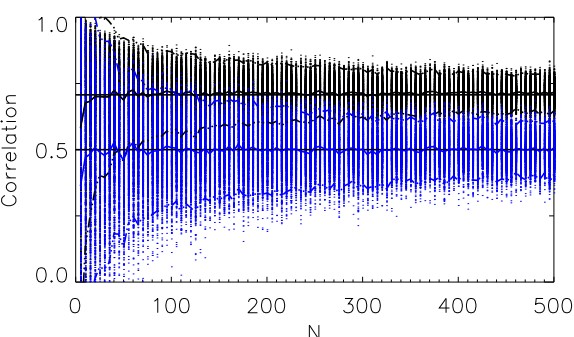

**Figure 6.** Correlations as function of dimension $N$. For each $N$ we draw 50 $N$-dimensional random vectors and calculate correlations of pairs of sample differences (blue, $corr(\mathbf{x}^k - \mathbf{x}^m, \mathbf{x}^l - \mathbf{x}^m)$) and correlations of sample differences and differences between ensemble mean and individual ensemble members (black, $corr(\mathbf{x}^k - \mathbf{x}^m, \overline{\mathbf{x}} - \mathbf{x}^m)$). The thick curves show the mean of the correlations, and the broken curves the mean $\pm$ two standard deviations In left panel each component of the vectors is drawn from a standard Gaussian, in right panel each component is drawn from a Gamma distribution (see caption to Fig. 4. The horizontal black lines indicates $1/2$ and $1/\sqrt{2}$.

Therefore, the distance between two sample members are a square-root of 2 larger than the distance between a sample member and the sample mean. The geometric interpretation is that the sample mean and any two sample members form an isosceles right triangle with the right angle at the sample mean (Hall et al., 2005; Palmer et al., 2006). The factor of $\sqrt{2}$ then comes from Pythagoras's equation. It is worth noting that the sample mean is special and is not drawn from the same distribution as the sample members. Considering the example of the high-dimensional unit cube from section 2a, the sample members would be located in the spikes while the sample mean would be close to the center.

Figure 4 demonstrates this in the simple situation where 50 N-dimensional vectors are drawn from prescribed distributions. The results are shown as function of $N$. When $N$ increases the spread of the distances decreases and for large $N$ the factor of $\sqrt{2}$ is clearly seen. This holds both for simple spherical Gaussian distributed vectors (left panel) and for Gamma distributed vectors with strong anisotropy in the components (right panel) although the convergence is faster in the Gaussian case.

Figure 5 shows the distances for AgERA daily mean precipitation for June (left panel) and for the monthly climatology of near-surface temperature for the CMIP5 ensemble (right panel). The distribution of the distances between sample members is shown together with the distribution of the distances between the sample members and the sample mean. The mean and width of these distributions are also shown in Table 2 both for these and the other data-sets. In all cases the factor of $\sqrt{2}$ are clearly seen for the mean values although the widths of the distributions are substantial in all cases. For the AgERA daily precipitation (Fig. 5 left) the two distributions are almost separated while this is not the case for the CMIP5 ensemble.

The indistinguishable interpretation claims that observations are drawn from the same distribution as the ensemble members. With this assumption and the considerations above, Christiansen (2018) explained the ubiquitous observation that the error (compared to observations) of the ensemble mean often is 30 % smaller $((1 - \sqrt{2})/\sqrt{2})$ than the typical error of the individual ensemble members (e.g., Gleckler et al., 2008). We also explained why the ensemble mean very often has a smaller error than all individual ensemble members (Christiansen, 2019).


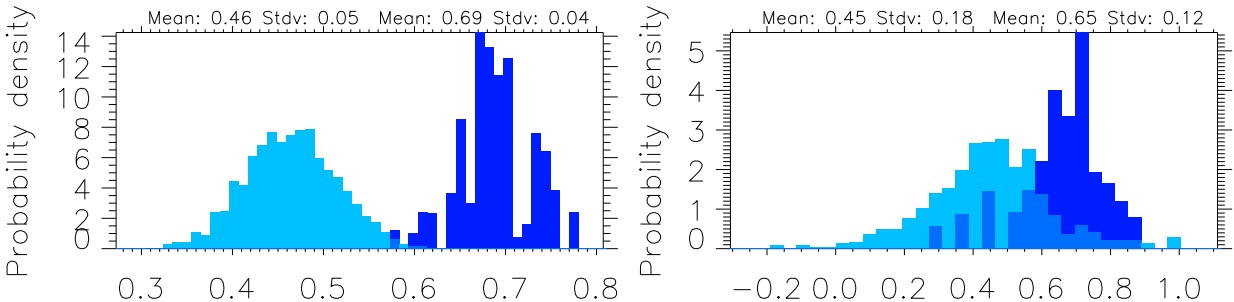

**Figure 7.** Correlations of pairs of sample differences $corr(\mathbf{x}^k - \mathbf{x}^m, \mathbf{x}^l - \mathbf{x}^m)$ (cyan) and correlations of sample differences and differences between sample mean and individual sample members $corr(\mathbf{x}^k - \mathbf{x}^m, \overline{\mathbf{x}} - \mathbf{x}^m)$ (blue). Left: Daily mean precipitation June from AgERA. Right: Monthly climatology of near-surface temperature in CMIP5.

The results in this subsection and subsections 4.2 and 4 do not only hold for the Euclidean (square) norm distance, but also
for, e.g., the maximum norm distance and the correlation distance ($\sqrt{1 - r^2}$, where $r$ is correlation).

## 4.3   Correlations between sample differences

Error correlations and correlations between model differences are important when studying the structure of a model ensemble and when comparing an ensemble to observations (Annan and Hargreaves, 2010; Pennell and Reichler, 2011; Bishop and Abramowitz, 2013).

We have in general $corr(\mathbf{x}^k, \mathbf{x}^l) = 1 - \frac{1}{2}||\hat{\mathbf{x}}^k - \hat{\mathbf{x}}^l||^2/N$, where $\hat{}$ indicates variables standardized to zero mean and unit variance. Therefore, with $\epsilon^k = \mathbf{x}^k - \mathbf{x}^m$ we have $\hat{\epsilon}^k = (\mathbf{x}^k - \mathbf{x}^m)/\sqrt{2}/\sigma$. We now get

$$corr(\mathbf{x}^k - \mathbf{x}^m, \mathbf{x}^l - \mathbf{x}^m) = corr(\epsilon^k, \epsilon^l) = 1 - \frac{1}{2}\left\|\hat{\epsilon}^k - \hat{\epsilon}^l\right\|^2/N = 1 - \frac{1}{2}\left\|\frac{\mathbf{x}^k - \mathbf{x}^l}{\sqrt{2}\sigma}\right\|^2/N = 1 - \frac{1}{2} = \frac{1}{2}, \tag{8}$$

where in the last step we have used Eq. 7. Thus, in high dimensions the correlation between sample differences is $1/2$.

Replacing $\mathbf{x}^l$ with the sample mean we get

$$corr(\mathbf{x}^k - \mathbf{x}^m, \overline{\mathbf{x}} - \mathbf{x}^m) = 1 - \frac{1}{2}\left\|\frac{\mathbf{x}^k - \mathbf{x}^m}{\sqrt{2}\sigma} - \frac{\overline{\mathbf{x}} - \mathbf{x}^m}{\sigma}\right\|^2/N = \tag{9}$$

$$1 - \frac{1}{2}\left\|\frac{\mathbf{x}^k - \overline{\mathbf{x}}}{\sqrt{2}\sigma} + (1 - 1/\sqrt{2})\frac{\mathbf{x}^m - \overline{\mathbf{x}}}{\sigma}\right\|^2/N = \tag{10}$$

$$1 - \frac{1}{2}(1/2 + (1 - 1/\sqrt{2})^2) = 1/\sqrt{2}. \tag{11}$$

In the last step we used the independence of the two terms and applied Eq. 6 to each.

Figure 6 shows the correlations for the simple random vectors as also used in Fig. 4. For all $N$ the correlations are distributed
around $1/2$ and $1/\sqrt{2}$. For small $N$ the spread is large but it decreases when $N$ increases and for large $N$ the correlations are very narrowly distributed around $1/2$ and $1/\sqrt{2} \approx 0.71$.





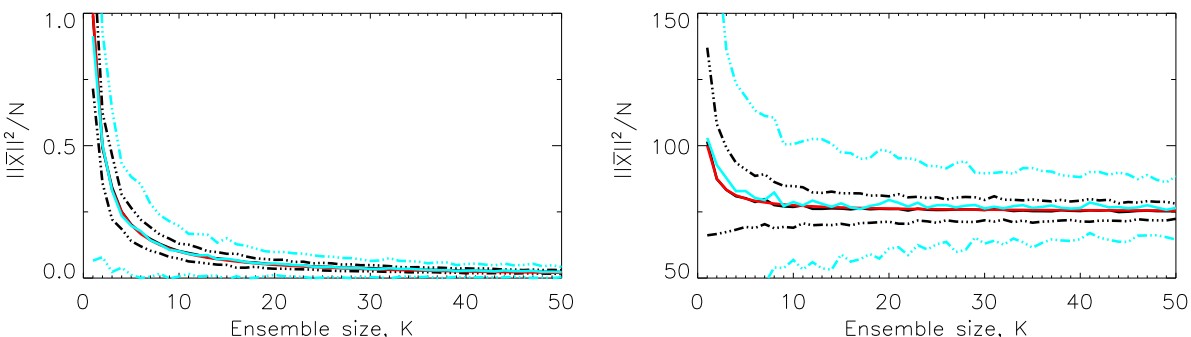

**Figure 8.** The length of the sample mean $||\overline{\mathbf{x}}||^2/N$ as function of sample size $K$. Black curves show results from $N = 100$ and blue curves for $N = 10$. For each $K$ results are based on 200 draws. The solid curves show the mean over these draws and the broken curves the mean $\pm$ two standard deviations. Red curve is analytic result (Eq. 12 with the theoretical values for $\mu$ and $\sigma$. In left panel each component of the vectors is drawn from a standard Gaussian $\mu^2 = 0$, $\sigma^2 = 1$, in right panel each component is drawn from a Gamma distribution, $\mu^2 = 75.00$, $\sigma^2 = 25.00$ (see caption to Fig. 4).

The correlations for AgERA daily mean precipitation for June and for the CMIP5 monthly climatology of near-surface temperate are shown in Fig. 7. The mean values are close to the high-dimensional values from Eqs. 8 and 9 although the spread is rather high. This is also the case for the other fields as reported in Table 2.

If we again assume that the observations are drawn from the same distribution as the ensemble members – the indistinguishable interpretation – the error correlation is $1/2$ (Eq. 8). On the other hand, if observations are near the ensemble mean – the truth centered interpretation– the error correlations will be zero as $\mathbf{x}^k - \overline{\mathbf{x}}$ and $\mathbf{x}^l - \overline{\mathbf{x}}$ are orthogonal. Error correlations around $1/2$ have been observed in many studies of climate models (e.g., Pennell and Reichler, 2011; Herger et al., 2018; Abramowitz et al., 2019), providing evidence for the indistinguishable interpretation.

With $\mathbf{x}^m$ replaced by observations, Eq. 9 gives the correlation between individual model errors and the model mean error. This quantity is shown in Figure 2 of Pennell and Reichler (2011) for the climatology of different variables in the CMIP3 multi-model ensemble and it is always close to $1/\sqrt{2} \approx 0.71$ as predicted by Eq. 9.

### 4.4    Effect of sample size

We now consider how the sample mean depends on the sample size. The ensemble mean is often used to estimate the forced re-
sponse from initial condition and multi-model ensembles (Frankcombe et al., 2018; Bengtsson and Hodges, 2019; Liang et al., 2020) and it is of interest to know how large an ensemble that is needed for the estimation to be saturated (Milinski et al., 2019).

     Letting $\overline{\mathbf{x}^\infty}$ represent the true (i.e., the distribution) mean of the sample, we get in the high dimensional case (reformulating Eq. 6)

$$||\overline{\mathbf{x}}||^2/N = \mu^2 + \sigma^2/K, \tag{12}$$





where $\mu^2 = ||\overline{\mathbf{x}^\infty}||^2/N$. Thus, $||\overline{\mathbf{x}}||^2$ converges like $1/K$ and the convergence is slowest where the sample spread is largest. Similar results have been presented by van Loon et al. (2007) and Potempski and Galmarini (2009) based on other arguments. See also Christiansen (2020) for the decay of the error of the ensemble mean when compared to observations.

The practical way to estimate the effect of sample size is to apply a bootstrap procedure to large sample of size $K^0$. From this sample we draw (with replacement) a number of sub-samples of size $K$, $K = 1, \ldots K^0$. From these sub-samples we calculate the mean and spread of $||\overline{\mathbf{x}}||^2/N$ for each $K$.

The mean is shown as function of $K$ – using the bootstrap procedure – in Fig. 8 for the simple examples with $N = 100$ and $N = 10$. For $N = 100$ (black curves) $||\overline{\mathbf{x}}||^2/N$ is narrowly distributed around the theoretical mean (Eq. 12) for both the Gaussian (left) and Gamma distributed samples (right). For $N = 10$ (cyan curves) $||\overline{\mathbf{x}}||^2/N$ is also distributed around the theoretical mean but with larger spread.

In the three previous subsections we studied the samples of daily June temperatures and of monthly climatologies. In the former the $N$-vectors consisted of spatial maps and in the latter of combined spatial climatologies for all 12 months. However, we also work in high-dimensionality when considering a single long time-series. The left panel in Fig. 9 shows time-series of the annual NH mean near-surface temperature for the MPI-GE 100 member initial condition ensemble and the CMIP5 45 member multi-model ensemble for the period 1961–2005. Both ensembles have been centered to their ensemble means in the first 10 years. Both ensemble means agree on a forced response consisting of an overall trend with some signals of volcanic eruptions after 1982 (El Chrichón) and 1991 (Mount Pinatubo). The spread of the multi-model ensemble is much larger than the spread of the initial condition ensemble.

The right panel shows $||\overline{\mathbf{x}}||^2/N$ as function of $K$. As expected from Eq. 12 the initial condition ensemble converges faster than the multi-model ensemble due to its smaller variance. Note the excellent agreement with Eq. 12 (red curves) where $\sigma^2$ has been estimated as the variance over time and all ensemble members and $\mu^2$ likewise estimated from the ensemble mean over all ensemble members. The large spread for the CMIP5 ensemble is due the well-known fact that the bias in global mean temperature is different for different models (Wang et al., 2014), which lead to a break-down of the condition of independence. This is not the case for the initial condition ensemble (see also Table 2). Smaller spread is obtained for the CMIP5 ensemble if each model is centered to its own (and not the ensemble) mean in the first 10 years.

## 5 Conclusions

It is well known that the number of samples necessary for a given coverage increases exponentially with the dimension. In this paper we have described other more non-intuitive properties of high-dimensional space such as the concentration of measures and waist concentration. In loose terms these properties state that independent sample members from the same distribution have the same lengths and that pairs of independent sample members are orthogonal. While most results are derived for iid random variables we discussed the extension to the non-iid situation and how the strength of the dependence is related to the effective dimension.


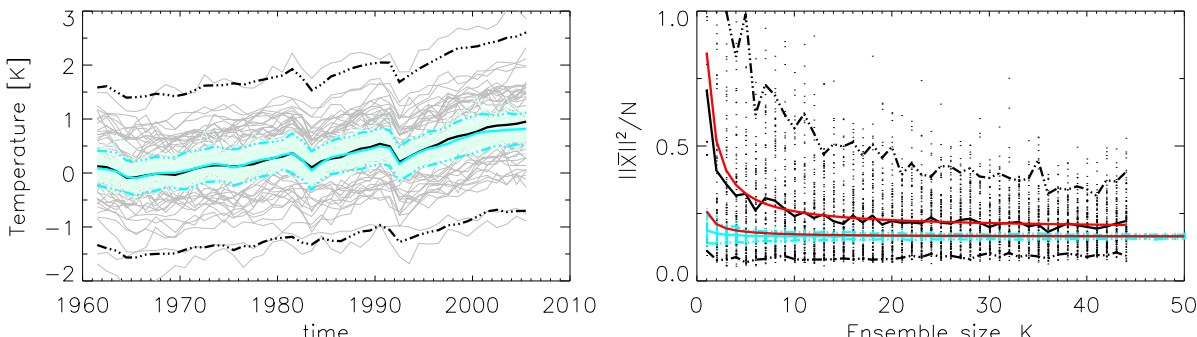

**Figure 9.** Left: Time-series of annual NH mean temperature from MPI-GE (black) and CMIP5 (cyan). Thick solid curves are ensemble means, dashed curves ensemble means $\pm$ two standard deviations, and thin curves are individual models. Each ensemble has been centered to its ensemble mean in the first 10 years. Right: The length of the ensemble mean $||\overline{\mathbf{x}}||^2/N$ as function of ensemble size $K$ for MPI-GE (black) and CMIP5 (cyan). The ensemble means $\pm$ two standard deviations are also shown. Theoretical results from Eq. 12 with $\sigma^2 = 0.094$, $\mu^2 = 0.164$ K$^2$ for MPI-GE and $\sigma^2 = 0.654$, $\mu^2 = 0.193$ K$^2$ for CMIP5 are shown in red.

We directly investigated to which extent these properties hold for typical climate fields and time-series. Ensemble modelling provides an obvious source of samples, but samples can also be obtained by considering, e.g., different days or years. We

330   investigated the monthly climatology of both an initial condition ensemble and a multi-model ensemble. We also investigated fields of daily means from a reanalysis. While the nominal dimensions of such fields are high, the effective dimensions are typically of the order 25-100, and it is not obvious to which degree the properties of high-dimensional dimension apply to such fields.

We found that for the global scale fields of near-surface temperature and precipitation both the concentration of measures

335   and the waist concentration hold to a reasonable degree. The lengths of the sample members are rather narrowly distributed around the mean length with widths (standard deviation) around 1/5–1/10 of the mean value. The angles between pairs of sample members are also rather narrowly distributed around $\pi/2$. This holds both when the samples consist of the climatology of different ensemble members from a model and when the samples consist of different daily means from a reanalysis.

Regarding the model ensembles, the concentration properties are better fulfilled for the initial condition ensemble (MPI-GE)

340   than for the multi-model ensemble (CMIP5). In the latter case the dependence of related models will result in these models being far from orthogonal.

Based on the concentration properties we derived simple analytical results that hold for large dimensions. These analytical results include: 1) The distances between two sample members are a factor of $\sqrt{2}$ larger than the distance between sample members and the sample mean. 2) The correlations between differences of pair of sample members are $1/2$ while the correlations between differences of sample members and the sample mean are $1/\sqrt{2}$. 3) An expression for how the sample mean depends on the sample size and on the sample spread. We found that these results describe the behaviour of the climate fields reasonable well.

We conclude that in many cases the concentration properties allow us a deeper understanding the behaviour of samples of climate fields. However, in each case it is important to investigate if the conditions of high dimensionality and independence are fulfilled. Even for global fields there is a substantial spread around the values predicted for the high dimensional limit.

We have only briefly mentioned the relation between observations and models. The relation depends on whether we assume that observations are drawn from the same distribution as the model ensemble (the indistinguishable interpretation) or whether we assume that the ensemble members are centered around the observations (truth centered interpretation). In the former case the results for individual model members also hold for observations as we discussed in section 4.2, while in the latter case results may be different. Many of the simple analytical results can be extended to situations where, e.g., the models are biased as explored in Christiansen (2020) using a simple statistical model that included both interpretations as limits.

*Competing interests.* The author declares that there is no conflict of interest.

*Acknowledgements.* This work is supported by the NordForsk-funded Nordic Centre of Excellence project (award 76654) Arctic Climate Predictions: Pathways to Resilient, Sustainable Societies (ARCPATH) and by the project European Climate Prediction System (EUCP) funded by the European Union under Horizon 2020 (grant agreement 776613).

We acknowledge the World Climate Research Programme's Working Group on Coupled Modelling, which is responsible for CMIP, and we thank the climate modeling groups for producing and making available their model output. For CMIP the U.S. Department of Energy's Program for Climate Model Diagnosis and Intercomparison provides coordinating support and led development of software infrastructure in partnership with the Global Organization for Earth System Science Portals.

The AgERA reanalysis was downloaded from https://cds.climate.copernicus.eu/.

The MPI Grand Ensemble Project https://www.mpimet.mpg.de/en/grand-ensemble/ was downloaded via EFGS from https://esgf-data.dkrz.de/projects/esgf-dkrz/.




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
