# Peer review of "The blessing of dimensionality for the analysis of climate data"

_Nonlinear Processes in Geophysics, 2021_

## Author Response (AR1)

**1 Review 1:**

2 Thanks for the positive review and the constructive comments.

Line 107: are the vectors a and b sampled from the Gaussian distribution mentioned above or are they generic?

The result is generic. I now cite Lehmann and Romano (2005) for a more general
 derivation.

Line 119: The inequalities in Eq. (4)-(5) very much resemble a large deviation principle. Making a reference to large deviations theory will strengthen the connection of high-dimensionality with the statistical mechanics theory presented in the paragraph starting in line 99.

In I have made the connection to 'large deviation theory' at the end of the section and included a few relevant references.

Line 185: When presenting the climate data, it would be very useful for the reader to know the dimensions N and sample size K in each case. Certainly, this is done later when analysing each case, but I believe doing it earlier could help the reading.

I now give the sample size when describing the climate data in section 4 and in Table 2. But the nominal dimension N depends on the details of the analysis, e.g., if we look at the monthly climatology or annual means, or the whole globe or Europe. Also, usually N is very large compared to the effective dimension  $N^*$ . Therefore, Nis given only in section 4.1 as before.

Line 228: The author mentions that  $x^k - \overline{x}$  is orthogonal to  $\overline{x}$  in high dimensions. This fact follows from the concentration of measures result presented in section 2.2, so I would encourage the author to make a clear reference to the theory.

25 I now briefly refer to the theory (waist concentration).

Line 240: The reference to the spikes of the unit cube is very helpful and in fact it illustrates the comment done in Line 228. The author might consider making this analogy earlier.

I now mention the sample mean and why it is special already in the description
 of the unit cube (170).

Clearly, the CMIP5 outputs deviated the most from having waist concentration, i.e. it provided non-orthogonal samples. This was attributed to high dependence. To what extent could it be attributed, instead, to low effective dimension of the climatological fields? From a personal perspective, I'd be interested in hearing the author's further views on this general question.

As I mentioned in my online response, I believe the negative skewness is due to dependence amongst models while the width of the bulk of the distribution is probably due to the effective dimension. The widths of the angles (for temperature) in Figs. 2 and 3 correspond roughly to an effective dimension around 25-50 (compare Fig. 1, right). This is also near the effective dimension found in near-surface temperature as mentioned in section 3 (l151).

The methods used to estimate the independent degrees of freedom usually assume that the samples are independent. If not the degrees of freedom will be underestimated. In our case it is the effective dimension of the climatology that will be

- 45 underestimated if the models are dependent. Likewise, if the number of indepen-
- 46 dent models were to be estimated, this number would be underestimated as the
- 47 climatologies are dependent.
- $_{48}$  I have added the sentence at line 218: A simple comparison between the distri-
- 49 butions of  $\phi$  in Figs. 2 and 3 with the distributions in Fig. 1 (from Gaussians) shows
- $_{\tt 50}$   $\,$  that the effective dimension is between 25-50 for temperature and several hundreds
- 51 for precipitation.
- 52 Technical Minor Corrections:
- 53 I have fixed the typos.

**54 Review 2**

55 Thanks for the positive review and the constructive comments.

l.85 & l.92: "a constant" I know this phraseology is used in related literature,
but I do not like it very much. I would have preferred to explicitly say something
like "... a constant for different realizations of the sampling process," or something
similar. Essentially this is a frequentists statistical argument, suggesting there is
a fixed distribution mean (the "constant") which can be approximated by a sample
mean.

I am not sure I really understand the reviewer's concern. As he mentions, 'a constant' is generally used in the literature and the meaning should be clear. So I have not made any changes here.

l.116: "sub-Gaussian" I am happy with this word, but it may be useful to include
a one- sentence definition of it (I am not sure how widely known the word is.)

I have briefly defined 'sub-Gaussian' as a distribution which tails decay at least as fast as the tails of a Gaussian distribution.

Section 3: the discussion of effective dimensionality is a well-worn topic in geo-69 physics, and it remains an important topic. It reminds me of a paper I wrote some 70 years ago (doi:10.1175/2007jas2298.1) on how suggested multimodality in a wave 71 amplitude index for the atmospheric circulation possibly is a statistical fluke: it 72 hinges on exactly the high dimensionality argument you are discussing here, but 73 interpreted slightly differently. In effect, standard statistic (moments) from a high 74 dimensional data set are always expected to exhibit these "blessing of dimensional-75 ity" properties, and more fancy, non-linear properties, such as multimodality, are 76 more-or-less by definition excluded. I think this is an important application of the 77 high dimensionality property, and I wonder whether you care to comment on it. 78

This is an interesting topic but I don't think I can contribute with much. When 79 it comes to using the central limit theorem to argue that an average is prone to be 80 Gaussian we should be careful. As I mention in 10.1175/2008JCLI2633.1 '... it has 81 been argued from the central limit theorem that the principal components must be 82 prone to Gaussianity because they are linear combinations of the local variables. 83 But, this argument suffers from the symmetry of the situation. Just as the PCs are 84 linear combinations of the local variables, the local variables are linear combinations 85 of the PCs ..' 86

I have moved the sentence about machine learning to the end of subsection 2.1 and in this connection I briefly mention that the 'concentration of measures' poses problems for clustering algorithms. A few references are added.

Section 4.2: I thought this section, despite its simplicity, was really thought provoking. I tried to interpret this in light of the well-known "signal-to-noise paradox"
Scaife et al. 2018 (https://doi.org/10.1038/s41612-018-0038-4), as it seems to be germane to that problem. Does the author agree that his discussion here may shed
light in the signal-to-noise problem? It would be a rather important application.

As I mentioned in my online response, I have had the same idea about the 'signalto-noise paradox' and I am in the process of including this in a coming paper. But I don't think this can explain why models underestimate the forced signal, which seems to be the key to the 'paradox'. 99 l.333 and Abstract: Perhaps I did not catch it but the dimensionality of 25-100 100 seems to fall a little bit from thin air. Can you please highlight where this estimate 101 is based on? Furthermore: Figs 2 & 3 show empirical distributions of  $\phi$ , which have 102 a given shape for independent data (Eq. 5). I would have thought that you can fit 103 a Gaussian to those distributions and thus estimate a value of N. Did you do this? 104 Does it give you the 25-100 estimate?

As mentioned in my online response there is actually a little more details and references regarding the number of degrees of freedom in the paragraph beginning at line 151.

The widths of the distributions of the angles are related to the degrees of freedom, but there is also an effect of the model dependence as mentioned in l214 that will disturb a direct calculation of the degrees of freedom.

111 A simple comparison between the distributions of  $\phi$  in Figs. 2 and 3 with the 112 distributions in Fig. 1 (from Gaussians) shows that the effective dimension is be-113 tween 25-50 for temperature and several hundreds for precipitation. This sentence 114 is now added to the manuscript near line 218.